High type I collagen density fails to increase breast cancer stem cell phenotype

Valadão Iuri C. 1
Ralph Ana Carolina L. 1
http://orcid.org/0000-0002-5114-1757 Bordeleau François 2
Dzik Luciana M. 1
http://orcid.org/0000-0001-8751-3412 Borbely Karen S.C. 3 4 5
Geraldo Murilo V. 6
Reinhart-King Cynthia A. 2
Freitas Vanessa M. 1 vfreitas@usp.br
1 Department of Cell and Developmental Biology, Institute of Biomedical Sciences, University of São Paulo , São Paulo , Brazil
2 Department of Biomedical Engineering, Vanderbilt University , Nashville, TN , USA
3 Department of Immunology, Institute of Biomedical Sciences, University of São Paulo , São Paulo , Brazil
4 Cell Biology Laboratory, Institute of Biological and Health Sciences, Federal University of Alagoas , Maceió , Brazil
5 Faculty of Nutrition, Federal University of Alagoas , Maceió , Brazil
6 Department of Structural and Functional Biology, Institute of Biology, University of Campinas , Campinas , Brazil
Singh Shree Ram
Electronic publication date: 2020 May 12
Publication date: 2020
Volume: 8
Electronic Location ID: e9153
Received 2020 Feb 11; Accepted 2020 Apr 18
Copyright: © 2020 Valadão et al.
Copyright year: 2020
Copyright holder: Valadão et al.
License: This is an open access article distributed under the terms of the Creative Commons Attribution License, which permits unrestricted use, distribution, reproduction and adaptation in any medium and for any purpose provided that it is properly attributed. For attribution, the original author(s), title, publication source (PeerJ) and either DOI or URL of the article must be cited.
License URL: https://creativecommons.org/licenses/by/4.0/

Keywords: Breast cancer, Cancer stem cells, Type I collagen, Mechanotransduction, Extracellular matrix

Funding: The State of São Paulo Research Foundation (FAPESP) 2015/02498-1 Brazilian National Council for Scientific and Technological Development (CNPq) 406683/2018-2 Brazilian National Council for Scientific and Technological Development (CNPq) 140078/2014-2 Santander International Mobility Scholarship This investigation was supported by The State of São Paulo Research Foundation (FAPESP grants 2015/02498-1) and from the Brazilian National Council for Scientific and Technological Development (CNPq grant 406683/2018-2). Iuri C Valadão was also supported by a scholarship from Brazilian National Council for Scientific and Technological Development (CNPq grant 140078/2014-2) and a Santander International Mobility Scholarship. The funders had no role in study design, data collection and analysis, decision to publish, or preparation of the manuscript.

==============================
Breast cancer is a highly frequent and lethal malignancy which metastasis and relapse frequently associates with the existence of breast cancer stem cells (CSCs). CSCs are undifferentiated, aggressive and highly resistant to therapy, with traits modulated by microenvironmental cells and the extracellular matrix (ECM), a biologically complex and dynamic structure composed mainly by type I collagen (Col-I). Col-I enrichment in the tumor-associated ECM leads to microenvironment stiffness and higher tumor aggressiveness and metastatic potential. While Col-I is also known to induce tumor stemness, it is unknown if such effect is dependent of Col-I density. To answer this question, we evaluated the stemness phenotype of MDA-MB-231 and MCF-7 human breast cancer cells cultured within gels of varying Col-I densities. High Col-I density increased CD44+CD24− breast cancer stem cell (BCSC) immunophenotype but failed to potentiate Col-I fiber alignment, cell self-renewal and clonogenicity in MDA-MB-231 cells. In MCF-7 cells, high Col-I density decreased total levels of variant CD44 (CD44v). Common to both cell types, high Col-I density induced neither markers related to CSC nor those related with mechanically-induced cell response. We conclude that high Col-I density per se is not sufficient to fully develop the BCSC phenotype.

Introduction

Breast cancer is the most frequent and the second deadliest cancer worldwide (Torre et al., 2015). Although the 5-year relative survival rate for breast cancer patients has recently reached almost 90% (Sant et al., 2015; Miller et al., 2016), incidence and death rates have been increasing in developing countries (Miller et al., 2016). Also, recurrence affects 30% of patients within 5 years after diagnosis and is related to factors such as hormone receptor status, disease stage, tumor differentiation (Cheng et al., 2012), and the existence of cancer stem cells (CSCs; Zhou et al., 2010). CSCs maintain intratumoral heterogeneity by being able to self-renew and to differentiate (Lapidot et al., 1994; Nassar & Blanpain, 2016). CSCs are believed to drive disease recurrence due to its enhanced resistance towards therapy, which is linked to a higher expression of drug transporters, higher resistance to DNA damage, and quiescence (Borst, 2012). Interestingly, low nuclear levels of proliferative marker Estrogen Receptor alpha (ERα) is common in BCSCs (Simões et al., 2011). Also, epithelial to mesenchymal transition (EMT), which includes downregulation of E-cadherin and upregulation of Snai1, is a process tightly correlated with the acquisition of stemness traits by breast cells (Mani et al., 2008).

Similar to its normal counterparts, CSCs are also identified by distinct immunophenotypes, such as the CD44+CD24− (Al-Hajj et al., 2003), and their characteristics are also maintained by the expression of pluripotency factors, such as Nanog (Jeter et al., 2009) and Oct-4 (Beltran et al., 2011). CSCs are modulated by the non-cellular and cellular components (Calabrese et al., 2007; Pickup, Mouw & Weaver, 2014) of the tumor microenvironment, the former mainly represented by the extracellular matrix (ECM).

The ECM is a highly dynamic structure that regulates a multitude of cellular processes, such as migration, invasion, proliferation, and differentiation (Pickup, Mouw & Weaver, 2014). The ECM is composed of a great variety of macromolecules, such as polysaccharides, glycoproteins, and proteins, with type I collagen (i.e., Col-I) being its most abundant component (Ricard-Blum, 2011). Increased Col-I density in the breast tumor-associated ECM potentiates tumorigenesis and metastasis (Provenzano et al., 2008; Kakkad et al., 2012). It also correlates with high stiffness (Fenner et al., 2014) in mammographically assessed high breast density (Alowami et al., 2003; Huo et al., 2015), which is per se one of the highest risk factors for breast cancer development (Li et al., 2005). In its turn, stiffness is a physical trigger that elicits a mechanically-induced cell response (i.e., mechanotransduction; Wang, 2017) by regulation of mechanotransducers, such as transcriptional coactivators YAP/TAZ (Dupont et al., 2011) Also, nuclear translocation of YAP/TAZ upregulates the BCSC phenotype both in vitro and in vivo (Cordenonsi et al., 2011; Kim et al., 2015a).

As Col-I enrichment fosters tumorigenesis, progression, and metastasis of breast cancer, we hypothesized that this effect relies on the upregulation of the BCSC phenotype. We tested this hypothesis by culturing MCF-7 and MDA-MB-231 human breast cancer cell lines within low-, intermediate- and high-density Col-I gels (Fig. 1). Cells were also cultured in two-dimensional (2D) surface of tissue culture plates and as mammospheres (MS) to respectively represent conventional and breast cancer stem cell (BCSC) culture (Fig. 1). We observed that high Col-I density upregulates CD44+CD24− BCSC immunophenotype and inhibits mesenchymal morphology of MDA-MB-231 cells. In MCF-7 cells, high Col-I density decreased the total levels of variant CD44 (CD44v). However, high Col-I density fails to upregulate CSC-related markers and clonogenic and mammosphere formation efficiency as well as matrix remodeling measured by Col-I fiber alignment. Overall, high Col-I density fails to upregulate the BCSC phenotype.

Figure 1 Scheme of cell culture environments used in this study.

Cell lines (MCF-7 and MDA-MB-231) were cultured on a 2D surface to represent conventional cell culture, while culture within floating Col-I gels was used to evaluate Col-I density effect on cells. Mammosphere culture was used to enrich cells with the BCSC phenotype.

Materials and Methods

Cell culture

Mammary tumor cell lines MDA-MB-231 and MCF-7 were acquired from the American Type Culture Collection (ATCC, Manassas, VA, USA). Human embryonic carcinoma cell line NTERA-2 was kindly provided by Professor Rodrigo A. Panepucci (Fundação Hemocentro de Ribeirão Preto, FUNDHERP, Brazil). MDA-MB-231, MCF-7, and NTERA-2 cells were cultured in DMEM-F12 (Sigma Aldrich, St. Louis, MO, USA) supplemented with 10% Fetal Bovine Serum (FBS; Cultilab, Campinas, Brazil) and 1% Penicillin/Streptomycin (Gibco, Langley, OK, USA). Cells were cultured in 2D surface of polystyrene tissue culture plates (Corning Inc., Corning, NY, USA) and within Col-I gels. Col-I gels were prepared on ice by mixing rat tail type I collagen (Cat. 354249; Corning Inc., Corning, NY, USA), 10X DMEM-F12, and 10X Reconstitution Buffer (2.2% NaHCO3, 4.8% HEPES in 0.2N NaOH) in a ratio of 8:1:1, respectively. In order to obtain different Col-I densities in the final mixture, the collagen solution was further diluted in 0.02N Acetic acid. Then, cell suspension was added to the mixture in a volume that would not exceed 1% of the final volume, to not significantly interfere with the final concentration of collagen. Col-I final mixture (175 µL) was added to wells of 24-well culture plates (Corning Inc., Corning, NY, USA) and placed in the CO2 incubator for 1 h to polymerize. Col-I gels were then covered with the appropriate cell culture media and were detached from the bottom of wells with the aid of a pipette tip (Fig. 1). We opted for floating gels to avoid cell attachment and spreading on the bottom of wells, which is hardly accomplished when using restrained gels.

Flow cytometry

Cells were plated on a 2D surface (1 × 104 cells/35 mm dish) and within Col-I gels (2 × 104 cells/gel) placed into 24-well plates (Corning Inc., Corning, NY, USA), as previously described. After 7 days in culture, Col-I gels were recovered, washed with PBS 1X, and incubated with 2.5 mg/mL type I collagenase (Cat. LS004196; Worthington Biochemical Corporation, Lakewood, NJ, USA) solution at 37 °C for 20 min under gentle agitation. Collagenase was used as it was shown to not alter the levels of CD44 expression (Biddle et al., 2013; Tsuji et al., 2017). Next, gel-recovered cells and cells in 2D were dissociated with 0.05% EDTA/PBS buffer and washed twice with PBS. Cells were incubated for 20 min in the dark at 4 °C with CD44-FITC (Cat. 555478; BD Biosciences, Franklin Lakes, NJ, USA) and CD24-PE (Cat. 555428; BD Biosciences, Franklin Lakes, NJ, USA) fluorophore-conjugated primary antibodies diluted 1:50 in blocking buffer (0.5% m/v BSA). After washing, cells were incubated with fixation buffer (0.5% m/v BSA in 2% m/v Paraformaldehyde) and analyzed on a BD FACSCalibur™ (BD Biosciences, Franklin Lakes, NJ, USA) flow cytometer. Unstained cells and magnetic beads (BD™ CompBead, EUA) stained with either CD44-FITC or CD24-PE alone were used to determine cell autofluorescence and to perform fluorescence compensation, respectively. At least 10,000 events were acquired in each run. Dot-plot graphs and manual gating (Fig. S1) were generated using FACSalyzer software, which is freely available online at https://sourceforge.net/projects/fcsalyzer/.

PCR array

MDA-MB-231 cells were cultured on a 2D surface and within Col-I gels for 7 days. Cells were recovered as previously described, and total RNA was extracted with RNeasy Mini kit (Cat. 74104; Qiagen, Hilden, Germany) according to the manufacturer’s instructions. The total RNA concentration was determined spectrophotometrically (Epoch; Biotek, Winooski, VT, USA) by measuring the 260/280 nm ratio, and total RNA integrity was checked by running an agarose gel electrophoresis. Next, cDNA was synthesized from 0.5 µg RNA using RT2 First Strand and mixed with RT2 SYBR Green mastermix according to manufacturer’s instructions. The final mixture was added to the wells of RT2 Profiler™ PCR Array Human Cancer Stem Cells (EUA Cat. 330231, Qiagen, Hilden, Germany) array plates. Briefly, this PCR array consists of a 96-well plate containing primers (1 pair/well) for amplification of CSC-related genes (84 wells), endogenous control genes (5 wells), and quality-check controls (7 wells). The reactions were read in StepOnePlus™ (ThermoFisher®, Waltham, MA, USA) quantitative PCR (qPCR) system according to the manufacturer’s recommendations for this equipment. Data were loaded in the analysis website (PCR Array Data Analysis Web Portal; Qiagen, Hilden, Germany), and expression of the 84 genes was normalized to the arithmetic mean of the expression of endogenous control genes. The online analysis platform also generated the heatmap of gene expression and gene clustering based on modulation of expression. A list with complete data is provided in Table S1.

qPCR

cDNA was obtained as described in the previous section and amplified with Power SYBR Green PCR Master Mix Kit (ThermoFisher®, Waltham, MA, USA) in a StepOnePlus™ (Applied Biosystems, Foster City, CA, USA) PCR system. Relative quantification of gene expression was calculated according to Pfaffl (Pfaffl, 2001) using GAPDH as an endogenous control for mRNA normalization. The sequences of primers are presented in Table S2.

Immunoblotting

Cells were cultured for 7 days on a 2D surface (5 × 103 cells/dish) or within Col-I gels (25 × 103 cells/gel), or 10 days in suspension for mammosphere generation (5 × 103 cells/well). Protein extraction was performed at 4 °C with ice-cold reagents. Cells in 2D surface were washed in PBS and then lysed with RIPA lysis buffer (150 mm Sodium Chloride, 1.0% NP-40, 0.5% Sodium Deoxycholate, 0.1% Sodium Dodecyl Sulfate in 50 mm Tris pH 8.0) containing phosphatase and protease inhibitor cocktail (Sigma, Ronkonkoma, NY, USA). Col-I gels were collected, washed with PBS, and briefly centrifuged for removal of excess liquid. Col-I gels were transferred to microtubes containing lysis buffer and triturated with the aid of a sharp point tip. Mammospheres (MS) were collected, washed in PBS, and lysed with RIPA buffer. Samples were kept under agitation for 20 min with occasional vortexing and then centrifuged (11,000 rpm for 20 min). Supernatant protein content was measured with Pierce™ BCA Protein Assay (ThermoFisher, Waltham, MA, USA) kit, except for gel lysates due to Col-I interference with the assay. Whole-cell lysates were then boiled (95 °C, 10 min) with β-mercaptoethanol for protein denaturation and electrophoresed (15–30 µg, 20–40% volume of Col-I lysate) in 10% polyacrylamide gels (SDS–PAGE). Proteins were transferred to Hybond ECL nitrocellulose membranes (GE, USA) and blocked overnight at 4 °C with TBS buffer containing 0.1% Tween-20 (TBS-T) and 5% non-fat milk. Blots were incubated overnight at 4 °C with the primary antibody in blocking buffer, washed with TBS-T (4 × 5 min), and incubated for 1 h at room temperature with secondary antibody in blocking buffer, followed by washing with TBS-T. Chemiluminescent detection was performed with Clarity Western ECL Substrate kit (BioRad, Hercules, CA, USA) in MF-Chemibis 3.2 (DNR Bio-Image Systems, Neve Yamin, Israel). Densitometry was performed in Fiji software (Schindelin et al., 2012). List of antibodies used for immunoblotting is provided in Table S3.

Analysis of self-renewal potential

Cells previously cultured on 2D surface (5 × 103 cells/plate) or within Col-I gels (25 × 103 cells/gel) for 7 days were collected, dissociated, and cell suspension was plated in ultra-low attachment 6-well plates (Corning Inc., Corning, NY, USA) at low cell density (5 × 103 cells/well) in DMEM-F12 containing 20 ng/ml bFGF (Sigma, Ronkonkoma, NY, USA), 20 ng/ml EGF (Sigma, Ronkonkoma, NY, USA) and 1X B-27 supplement (Gibco, Langley, OK, USA). Plates remained undisturbed in a CO2 incubator for 10 days, and then 25 images were acquired per well in an Axio Vert.A1 microscope equipped with an AxioCam MRc camera (Carl Zeiss, Oberkochen, Germany). Mammospheres were segmented in Fiji by sequentially applying minimum and median filters of 1- and 3-pixel radius followed by an auto threshold.

Immunofluorescence

Cells were cultured on a 2D surface (100 cells/13 mm round glass coverslips) (Knittel, Fellbach, Germany) for 7 days, then washed with PBS, fixed with 4% Paraformaldehyde (PF) for 10 min, washed once again and permeabilized with 0.2% Triton X-100 in PBS. Next, cells were incubated for 1 h in blocking buffer containing 1% BSA, 0.2% Triton X-100, 0.05% sodium azide, and 10% Goat Serum (KPL Inc., Gaithersburg, MD, USA) in PBS. Cells were incubated overnight at 4 °C with primary antibody diluted in blocking buffer, washed with PBS containing 0.05% Tween-20 (PBS-T) and incubated for 1 h at room temperature with secondary antibody in blocking buffer. Cells were washed with PBS, followed by distilled water, and coverslips were mounted on glass slides with SlowFade® Gold or ProLong® Gold mounting medium with DAPI (ThermoFisher, Waltham, MA, USA). Col-I gels were washed, fixed with 4% PF for 15 min, and incubated with 0.15 M Glycine in PBS for 10 min to quench aldehyde groups. Next, gels were washed, permeabilized with 0.5% TX-100 in PBS, and blocked overnight at 4 °C. Gels were then cut in 4–6 pieces, and each piece was incubated overnight at 4 °C with a primary antibody. Gels were extensively washed with PBS-T and incubated for 1 h at room temperature with appropriate secondary antibodies. After extensive washing with PBS-T and water, gels were mounted on glass slides and covered with either SlowFade® Gold or ProLong® Gold mounting medium with DAPI (ThermoFisher, Waltham, MA, USA). MS were collected in microtubes previously washed with 0.5% BSA in PBS. After pelleting, MS were suspended with PBS, and small aliquots (~25 µL) were pipetted in areas of glass slides delimited with a hydrophobic pen (Super PAP Pen; ThermoFisher, Waltham, MA, USA). Glass slides were air-dried for 5 min at 37 °C in an incubator, fixed for 10 min and incubated with 0.15M Glycine in PBS. After permeabilization (0.5% TX-100 in PBS) for 15 min, slides were washed with PBS and incubated with a blocking buffer. Next, slides were incubated overnight at 4 °C with primary antibodies, washed with PBS-T and incubated with secondary antibodies for 1 h. Slides were further washed with PBS-T and MS were covered with SlowFade® Gold or ProLong® Gold mounting medium with DAPI (ThermoFisher, Waltham, MA, USA), followed by a 13 mm round glass coverslip. Samples were rather stained with SYTOX Green (Molecular Probes, USA; 1:5,000 dilution) instead of DAPI when imaging was performed with Axio Observer.Z1. A list of antibodies used for immunofluorescence is provided in Table S3.

Immunofluorescence image analysis

Images were acquired on an Axio Vert.A1 fluorescence microscope or Axio Observer.Z1 with a motorized stage operated by Zen Blue 2.0 software and located at CEFAP (University of São Paulo, Brazil). Z-stacks were deconvolved with Richardson-Lucy Total Variation (RLTV) algorithm in the DeconvolutionLab2 plugin (EPFL, Lausanne, Switzerland) of Fiji. Cell nuclei were segmented manually by Gaussian blur filter (3-pixel radius) application followed by auto threshold and watershed partitioning. Mean fluorescence intensity (MFI) of nuclear markers was calculated as the fluorescence intensity of a nuclear marker per cell nucleus area. For cell membrane markers, the “Moments” threshold was applied after an image was subjected to background subtraction (rolling ball radius = 5-pixel radius) and minimum filter (0-pixel radius).

Clonogenic assay

Cells previously cultured on a 2D surface (5 × 103 cells/plate) or within Col-I gels (25 × 103 cells/gel) for 7 days were collected, dissociated and the cell suspension was cultured in 60 mm culture plates at low cell density (250 cells/plate) for 14 days, with cell media exchange every other day. Cells were washed with PBS, fixed with 4% PF for 10 min, and stained with 0.1% Crystal Violet solution in 20% Methanol for 2 min. Cells were cautiously washed with distilled water and air-dried overnight at room temperature. Digital images of the colonies were acquired in a BioDoc-It (UVP Inc., San Gabriel, CA, USA) imaging system and counted in Fiji software following instructions of a previous report (Cai et al., 2011).

Live cell imaging and confocal reflectance

Cells were cultured within Col-I gels (250 cells/gel) for 7 days in 24-well glass-bottom culture plates (MatTek, Ashland, MA, USA). Staining with CellTracker™ Green (ThermoFisher, Waltham, MA, USA) was performed as previously described (Ueda et al., 2004). Briefly, gels were washed twice with serum-free media and incubated for 45 min with 25 µm CellTracker™ Green (ThermoFisher, Waltham, MA, USA) in a CO2 incubator. Next, gels were washed with serum-free media and incubated with complete media in a CO2 incubator until analysis. Confocal reflectance microscopy was performed as described (Kraning-Rush et al., 2012). Briefly, images were acquired on an Axio Observer.Z1 inverted microscope with LSM 700 (Carl Zeiss, Oberkochen, Germany) in a 5% CO2 atmosphere and 37 °C humidified environment. Samples were illuminated through an 80/20 dichroic mirror with 405 nm laser and a LD- C-Apochromat 40x/1.1 NA water immersion objective. Cells were imaged with a 488 nm laser. Z-stacks (0.31 µm × 0.31 µm × 0.95 µm; xyz) were acquired in the center/middle of the gels. Cells were later manually segmented in Fiji.

Angular distribution of Col-I fibers

We analyzed the angular orientation of the Col-I fibers according to a previous study (Franco-Barraza et al., 2016). Briefly, the images were opened in Fiji and processed in the OrientationJ plugin (Rezakhaniha et al., 2012). “Direction” function was used to determine the most frequent fiber angle in the image (i.e., mode). “Distribution” function was then used to generate a histogram of the number of fibers (y-axis) at each orientation angle between −90 ° and +90° (x-axis). Next, the “Analysis” function was used to assess the hue of fiber orientation angles, and 0° was assigned with cyan and normalized to the mode angle in Photoshop CC software (Adobe Systems, San Jose, CA, USA).

Degree of Col-I fiber alignment

The anisotropy (directional dependance) of Col-I fibers was evaluated as previously described (Wang et al., 2018). Briefly, the angular distribution of the fibers was approximated to a normal (Gaussian) distribution using GraphPad Prism v.7 software (GraphPad Software, San Diego, CA, USA). The anisotropy ratio, which quantifies the directionality of orientation, was calculated as the ratio between the maximum height and Full Width at Half Maximum (FWHM) of the Gaussian distribution. The ratio between fibers within 30° from the mode angle (i.e., 15° to +15°) and total fibers was also used to access coherency in angle orientation (Franco-Barraza et al., 2016).

Statistical analysis

GraphPad Prism v.7 was used for statistical analysis. Means of three or more groups of an independent variable were compared by one-way ANOVA followed by Bonferroni’s post-test. Means of three or more groups of two independent variables were compared by two-way ANOVA followed by Tukey’s post-test. The difference between groups was considered statistically significant when p < 0.05. The absence of asterisks in graphics means the absence of a statistically significant difference.

Results

Effect of Col-I density on CSC immunophenotype of mammary cancer cells

We initially evaluated by flow cytometry the levels of the well-known CSC immunophenotype CD44+CD24− in MDA-MB-231 and MCF-7 human breast cancer cell lines grown on a 2D surface or within Col-I gels. Both cell lines presented some level of CD44+CD24− immunophenotype (Fig. 2A). MCF-7 cells showed similar levels of CD44+CD24− subpopulation in all groups, while MDA-MB-231 cells showed a higher proportion of CD44+CD24− subpopulation in gels of intermediate and high Col-I density (Fig. 2B). These results suggest that the increase in Col-I density favors the CSC phenotype in MDA-MB-231 cells.

Figure 2 Flow cytometry-based analysis of CD44+CD24− subpopulation levels in breast cancer cell lines cultured on a 2D surface and within Col-I gels.

Cell lines (MCF-7 and MDA-MB-231) were cultured for 7 days on a 2D surface and within Col-I gels, then collected, fixed and immunolabeled with CD44-FITC and CD24-PE. (A) Dot-plot graphs with subpopulations expressing none, one or both the analyzed molecules. The lower-right gate (Q3) contains the subpopulation of interest (CD44+ CD24−). Each cell line has a specific autofluorescence intensity, resulting in an equally specific division of the graphs. (B and C) Bar graphs represent the mean ± SEM of at least three independent experiments. (One-way ANOVA followed by Bonferroni post-test, *p < 0.05).

Effect of Col-I density on the CSC-related genes in MDA-MB-231 cells

High throughput techniques allow concomitant evaluation of the various molecular components that regulate the functional properties of CSC. We then evaluated the expression of dozens of genes related to BCSC phenotype in MDA-MB-231 cells by a PCR array kit (RT2 Profiler; Qiagen, Hilden, Germany). Col-I modulated the expression of 29 CSC-related genes, relative to the traditional monolayer culture. Of these 29 genes, 13 were downregulated, and 16 were upregulated (Fig. 3A). Positively regulated genes included classic pluripotency markers (Nanog, Lin28a, Pou5f1/Oct4).

Figure 3 Gene expression analysis of markers associated with the CSC phenotype of MDA-MB-231 cells.

(A) Hierarchical clustering of 29 differentially expressed genes in MDA-MB-231 cells cultured on a 2D surface or within Col-I gels. The increasing subdivision of the initial branch (from left to right) approximates genes with increasing similarity of expression modulation. The heatmap represents a single independent experiment. The magnitude of gene expression is represented by color scale, from blue (smaller magnitude) to red (larger magnitude). (B) RT-qPCR analysis of Nanog, Pou5f1/Oct4 and Snai1 mRNA expression. Results are presented in log2 fold change expression relative to 2D surface group after normalization with Gapdh mRNA. Bar graphs represent the mean ± SD of three biological replicates (One-way ANOVA followed by Bonferroni post-test; *p ≤ 0.05 and **p ≤ 0.01, relative to 2D surface group).

Nevertheless, Col-I densities similarly modulated gene expression (Fig. 3A). In RT-qPCR validation, cells grown within Col-I gels showed higher Nanog levels compared to those grown on monolayer, while Pou5f1/Oct4 levels remained unaltered and Snai1 levels were diminished in low Col-I density. Once again, Col-I density did not modulate the expression of analyzed genes (Fig. 3B). These results suggest Col-I modulates the expression of CSC-related genes relative to monolayer culture regardless of its density.

Effect of Col-I density on CSC-related protein levels in MDA-MB-231 cells

Protein levels of CSC-markers were first evaluated by immunoblotting. Nanog levels were undetectable in cells cultured either in monolayer or within Col-I gels, while CD44 and Integrin-α6 levels were similar among all groups (Figs. 4A and 4B). In contrast, monolayer cells presented high levels of YAP and its positive regulator PP2A in comparison to cells cultured within Col-I gels. The levels of an inhibitor and inducer of YAP activity, namely Sav1 and Wbp2, remained unaltered (Figs. 4A and 4B). Importantly, Col-I density did not modulate the protein levels of the proteins, as mentioned above. We next evaluated the subcellular localization of CSC-related markers by immunofluorescence. Both Nanog (Figs. 5A–5D) and YAP (Figs. 5E–5H) were mainly cytoplasmic, and as expected, CD44 was located at the cell membrane (Figs. 5I–5L). Notably, Col-I density modulated neither subcellular localization nor levels of CD44 (Fig. 5P) and nuclear YAP (Fig. 5Q) nor proliferation measured by PCNA (Fig. 5R).

Figure 4 Immunoblotting of CSC-related and mechanotransduction markers from MDA-MB-231 cells cultured on a 2D surface or within Col-I gels.

(A) Representative gels of CSC-related and cell mechanotransduction markers. Whole cell lysate of NTERA-2 human embryonic carcinoma cell line was used as a positive control for the expression of Nanog. β-actin was used as endogenous control. (B) Bar graphs represent mean ± SD of at least two independent experiments (One-way ANOVA followed by BonFerroni post-test; **p < 0.01, relative to the 2D surface group of the respective POI).

Figure 5 Immunofluorescence of CSC-related, mechanotransduction and proliferation markers from MDA-MB-231 cells cultured on a 2D surface or within Col-I gels.

(A–O) Representative images of the expression of CSC-related (Nanog and CD44), mechanotransduction (YAP) and proliferation (PCNA) markers in gray scale. Cell nuclei were replaced by their contours (white dashed line) for better visualization of Nanog and YAP nuclear expression. Scale bar: 20 μm. Box & Whisker’s graph represents median and 5th/95th percentiles, as well as mean (+) of Mean Fluorescence Intensity (MFI) for YAP (P) and CD44 (Q) markers. (R) Bar graph represents mean percent ± SD of PCNA+ cells. Experiments were performed independently at least twice, with number of evaluated cells (n) ranging from 50 to 800. Objectives used: (A–H) 63× (2D) and 40× (Col-I); (I–L) 63×; (M–O) 10×.

Effect of Col-I density on the CSC-related functional properties of MDA-MB-231 cells

We proceeded to the functional analysis of MDA-MB-231 cells cultured within gels of varying Col-I densities. We evaluated the colony formation efficiency of MDA-MB-231 cells by the clonogenic assay in tissue culture plates, and the self-renewal capability by the mammosphere formation assay in ultra-low attachment plates. Both properties remained unaltered among groups, whether comparing the culture environment (monolayer vs. Col-I) or Col-I density (Figs. 6A–6J). Mammosphere area, diameter, and circularity were also very similar among groups (Figs. 6K–6M). These results suggest Col-I density modulates neither proliferation nor self-renewal of MDA-MB-231 cells.

Figure 6 Clonogenic and self-renewal potential of MDA-MB-231 cells cultured on a 2D surface or within Col-I gels.

(A–D) Evaluation of the clonogenic potential of MDA-MB-231 cells previously cultured on a 2D surface or within Col-I gels. Cells were collected and processed to obtain individualized cell suspension. After plating (500 cells/35 mm plate), cells were cultured for 14 days and then washed, fixed and stained with crystal violet. The images were acquired with a transilluminator and represent technical replicates of one independent experiment. (E) Bar graphs represent mean ± SD of five independent experiments performed in technical quadruplicates. (F–I) Evaluation of mammosphere (MS)-formation efficiency by MDA-MB-231 cells previously cultured on a 2D surface or within Col-I gels. Cells were collected, processed to obtain individualized cell suspension and grown in ultra-low attachment 6-well plates (5,000 cells/well) with culture medium enriched for MS formation. After 10 days, images of at least 25 fields per well were acquired. Dashed squares outline zoomed-in MS, which have a minimum diameter of 50 μm (scale) and compact shape. Bar graphs (J) and scatter plots (K–M) represent mean ± SD of five independent experiments, performed in technical triplicate.

Morphological characterization of Col-I gels and MDA-MB-231 cells

Col-I gel architecture is affected by factors such as cell culture media, gel thickness, and cell presence, while cell morphology also changes in response to Col-I density (Carey et al., 2012). We then evaluated the morphology of acellular and cellular Col-I gels and the associated cells. To that end, gels containing live cells were stained with Cell Tracker Green (ThermoFisher, Waltham, MA, USA) and imaged in confocal microscopy (Fig. S2). We observed that the width (Fig. 7A), length (Fig. 7B), and straightness (Fig. 7C) of Col-I fibers remained unaltered, regardless of Col-I density or cell presence. The porous area of acellular Col-I gels decreased with the increase in Col-I density (Fig. 7D). High Col-I density impaired cell elongation measured by cell aspect ratio (i.e., cell major axis/cell minor axis; Fig. 7G). Col-I density did not modulate cell perimeter (Fig. 7E), area (Fig. 7F), solidity (Fig. 7H), or circularity (Fig. 7I). These results suggest Col-I density minimally alters the morphology of both cells and Col-I fibers.

Figure 7 Analysis of individual shape descriptors of Col-I fibers and MDA-MB-231 cells.

Col-I fibers were imaged by confocal reflectance while cells were stained with CellTracker Green right before image acquisition. (A–D) Bar graphs represent mean ± SD of the width (A), length (B), straightness (C) of Col-I fibers and porous (non-fibrillar) area of the gels (D). The “−” and “+” signs (A–C) identify acellular and cellularized gels, respectively. (E–I) Column scatter plots of the perimeter (E), area (F), aspect ratio (G), solidity (H) and circularity (I) of evaluated MDA-MB-231 cells. Aspect ratio, solidity and circularity values are represented in arbitrary unities (a.u.). Column scatter plots represent mean ± SD of at least 20 cells imaged in three independent experiments. Statistical analysis was performed using one-way (D, E–I) or two-way ANOVA (A–C) followed by Bonferroni’s post-test (D, G; *p < 0.05, relative to LD).

Col-I fiber organization in cellular and acellular gels

Our next goal was to analyze the microarchitecture of Col-I gels and its interaction with MDA-MB-231 cells. Gels stained with Cell Tracker Green were imaged in fluorescence and reflectance mode on a confocal microscope to detect cells and Col-I fibers, respectively. Cells did not show a preferential orientation (Figs. 8A–8C), and Col-I fibers were similarly distributed across the gel, regardless of cell presence (Figs. 8D–8I). Normalized angular orientation visually demonstrated a similar variety of angle distribution among groups (Figs. 8J–8O). We next quantitatively analyzed Col-I fiber orientation by measuring the proportion of fibers within 30° of the most observed angular orientation (i.e., mode). We observed no difference between groups (Figs. 9A and 9B) regardless of cell presence and Col-I density, implying these did not modulate Col-I fiber angular orientation. We further evaluated the global angular orientation of Col-I gels by calculating their anisotropy ratio values, which quantifies the directionality of fiber angular orientation. To do so, we first subjected Col-I fiber frequency histograms (Fig. 9A) to a Gaussian fit (Fig. 9C), and the anisotropy ratio was later calculated as the ratio between Gaussian fit maximum height by its full width half maximum (FWHM). Anisotropy ratio of gels was similar regardless of cell presence or Col-I density (Fig. 9E), further suggesting such parameters did not affect global fiber angular orientation.

Figure 8 Microstructure analysis of Col-I fibers by confocal reflectance.

MDA-MB-231 cells were stained with CellTracker Green® and imaged alive with a confocal microscope. Simultaneously, Col-I fibers were imaged by confocal reflectance microscopy. (A–C) Images depict cells (green) interacting with Col-I fibers (gray scale) in gels of varying Col-I densities. Images represent maximum intensity projections of z-stacks. (D–I) Maximum intensity projections of light reflected by Col-I fibers, by means of confocal reflectance technique. (J–O) Angular normalization of Col-I fibers detected in (D–I), with the most recurrent angle (i.e., mode) being normalized to 0° and attributed to cyan color. Angle assignment to each fiber was performed by OrientationJ plugin in Fiji software, while hue normalization of the color scale was performed in Adobe Photoshop CC software. Scale bar = 25 μm.

Figure 9 Col-I fiber alignment in acellular and cellularized Col-I gels.

(A) Histogram represent normalized mean Col-I fiber frequency per angular orientation of gels. Vertical dashed lines highlight fiber angle proportion ranging 30° from mode angle (i.e., 0°) and quantified in (B). (B) Box & whiskers graphs represent median, 95th/5th percentiles and mean (+) of Col-I fiber proportion between +15° and −15° relative to mode angle. Images (n = 20–26) were acquired from three independent experiments. (C) Histogram represent Gaussian fit of normalized mean Col-I fiber frequency per angular orientation of gels. (D) Representation of Maximum Height and Full Width at Half Maximum (FWHM) of a Gaussian fit. (E) Bar graphs represent mean anysotropy ratio ± SD of three independent experiments, with at least 20 images analyzed per group.

Effect of Col-I density on CSC markers of mammosphere-derived MCF-7 cells

Previous results showed a minimal positive effect of Col-I density on the CSC phenotype of MDA-MB-231 cells. We then tested whether Col-I density associated with a known stimulating environment for CSC development and maintenance would favor such a process. To do so, MCF-7 mammospheres were dissociated into single cells that were cultured within Col-I gels and fed with mammosphere specific media for a week (Fig. 10A). Gels were subjected to immunofluorescence or protein extraction for immunoblotting. Col-I did not significantly affect Integrin-α6 and CD44 standard (CD44s) levels (Figs. 10B and 10D). However, the increase in Col-I density reduced CD44 variant (CD44v) and E-cadherin and increased Sox2 levels (Figs. 10B and 10D). In contrast, high Col-I density upregulated total levels of estrogen receptor α (ERα). Once again, Col-I density failed to regulate total YAP levels (Figs. 10C and 10D). Regarding the subcellular localization of CSC markers, we observed a subtle cytoplasmic expression of Sox2 in mammospheres but no detectable expression in cells cultured within Col-I gels (Figs. 11A–11D). YAP was localized throughout the whole cell (Figs. 11E–11H), and its levels were similar regardless of Col-I density (Fig. 11Y), as similarly noted for ERα (Figs. 11I–11L and 11X). Both CD44 (Figs. 11M–11P) and E-cadherin (Figs. 11Q–11T) were localized at the cell membrane, and their levels were similar among groups (Figs. 11Z and 11AA). The levels of proliferative cells identified by Ki67 nuclear expression (Figs. 11U–11W) were also unaltered by Col-I density (Fig. 11BB). Overall, Col-I density failed to modulate CSC markers levels and subcellular localization in mammosphere-derived MCF-7 cells.

Figure 10 Immunoblotting of CSC-related and mechanotransduction markers of MS-derived MCF-7 cells cultured within Col-I gels.

(A) MCF-7 cells were plated for 10 days in ultra-low attachment plates for MS generation. Then, MS were collected, dissociated, and plated into Col-I gels. After 7 days, Col-I gels were lysed for protein extraction. Representative gels for detection of CSC-related (B) and mechanotransduction markers (C) in MS and Col-I gels whole-cell lysates. The NTERA2 human embryonic carcinoma cell line was used as a positive control for Sox2 expression. (D) Densitometric analysis of immunoblottings. Bar graphs represent mean ± SD of at least two independent experiments.

Figure 11 Immunofluorescence of CSC-related, mechanotransduction and proliferation markers in MS-derived MCF-7 cultured within Col-I gels.

(A–W) Representative images of the expression of molecular markers (gray scale) and cell nuclei (blue). Cell nuclei were replaced by their contours (white dashed line) for better visualization of YAP (E–H) and ERα nuclear expression (I–L). Scale bar: 20 μm. (X–AA) Box & Whiskers plot represent Mean Fluorescence Intensity (MFI) of the evaluated markers. Plots represent median, 95th/5th percentiles and mean (+) of each group data. (BB) Bar graph represents mean percentage (± SD) of cells expressing Ki-67 proliferation marker. Data refer to cells (n = 50–200) from at least two independent experiments.

Discussion

ECM is an important modulator of several physiological and pathological processes. Moreover, its structural composition is extremely varied and complex (Bonnans, Chou & Werb, 2014). Understanding how collagen enrichment in the mammary gland plays its significant role in breast cancer is thus mandatory. On the other side, CSC is a focus of attention due to their high resilience and contribution to tumor onset and maintenance (Nassar & Blanpain, 2016). To further shed light on this topic, we evaluated whether high Col-I density positively regulates stemness in breast cancer cells.

The CD44+CD24− subpopulation was the first identified as CSC in breast cancer (Al-Hajj et al., 2003). Our results show no difference in CD44+CD24− subpopulation levels between cells cultured in 2D surface or within Col-I gels. Nonetheless, CD44+CD24− subpopulation was increased in HD relative to LD condition. However, a recent study has shown that culture in Col-I scaffolds induces higher levels of CD44+CD24− subpopulation than 2D surface (Chen et al., 2012). The Col-I framework difference between this study and ours may have a role in such disparity (Tan et al., 2015) and should be a subject of further research.

Pluripotency markers are crucial for the maintenance of CSC-related phenotype and functional activity (Jeter et al., 2009; Stolzenburg et al., 2012; Tang et al., 2015). We observed that high Col-I density upregulates Nanog but not Oct4 transcripts relative to cell culture in 2D surface, as seen before (Reynolds et al., 2017). Our study did not address whether this effect is due to Col-I per se or to the distinct culture environments of 2D surface and Col-I gels (3D). Further studies comparing CSC-related genes of cells cultured in 2D and Col-I-coated 2D are then needed. High Col-I density modestly impacts Nanog nuclear levels in colorectal carcinoma cells in comparison to low Col-I density (Pankova et al., 2019). As for MCF-7, our results are in line with Sox2 and Bmi1 transcripts, but not nuclear ERα, being increased in Col-I-enriched ER+ murine breast tumors (Shea et al., 2018).

A multitude of cell surface proteins and glycoproteins have been associated with the BCSC phenotype. In contrast to our results, CD44 is highly expressed in Col-I-enriched regions of human breast tumors (Pang et al., 2016). Of note, tumor-associated macrophages (TAM; Esbona et al., 2018) and CAFs (Sharon et al., 2015) are enriched in desmoplasia and are important source of Osteopontin, a glycoprotein that both interacts with CD44 and increases its levels (Rao et al., 2013; Sharon et al., 2015). Thus, the absence of stromal cells in our experimental model may explain the lack of measurable increase in CD44 levels similar to what was observed in those cancer-associated desmoplasias. We also observe that high Col-I density downregulates CD44v levels without a compensatory increase in CD44s levels in MCF-7 cells. CD44v+ tumor cells are highly metastatic (Hu et al., 2017), while CD44s mediates invadopodia activity and maintenance of BCSC traits (Zhao et al., 2016; Zhang et al., 2019). Further studies are necessary to evaluate the impact of CD44v decrease in MCF-7 phenotype. Integrin-α6 preferentially binds laminin and is an important marker of various stem cell types (Krebsbach & Villa-Diaz, 2017). Indeed, Integrin-α6 levels are upregulated by increasing Col-I density of Col-I/laminin-rich ECM (e.g., Matrigel) gels (Paszek et al., 2005). The increase in Col-I density alone fails to upregulate Integrin-α6 levels in our experimental model. Thus, laminin may have a critical role in Col-I density-mediated Integrin-α6 upregulation. Lastly, our results suggest that previously reported downregulation of E-cadherin by Col-I (Cheng & Leung, 2011) relies on Col-I density.

Proliferation markers may also aid in the identification of BCSC. For example, ALDH+ breast tumors display higher proportion of proliferation marker PCNA and a worse clinical outcome (Gong et al., 2010) while Ki-67+, another classical proliferation marker, is essential for the functional characteristics of BCSC (Cidado et al., 2016). We demonstrate that Col-I density does not modulate Ki67 levels of MCF-7 cells, which corroborates in vitro and in vivo findings of similar studies also involving ERα+ environment (Barcus et al., 2015, 2017; Shea et al., 2018). In contrast, an increase in Col-I density appears to reduce PCNA levels in MDA-MB-231 cells and corroborate in situ proliferation downregulation previously shown (Kim et al., 2015b). In conclusion, cells with different ER status may respond differently to Col-I density concerning the expression of cell proliferation markers.

Mechanical factors are also critical in modulating biological processes. The Hippo pathway mediates the cellular response to these mechanical factors (Cordenonsi et al., 2011). In our study model, Col-I density fails to modulate YAP mechanotransducer levels or subcellular localization in MDA-MB-231 and MCF-7 cells. On the other hand, YAP response to Col-I density varies greatly even with the phenotypically indistinguishable fibroblasts and mesenchymal stem cells (Ali, Chuang & Saif, 2014; Ishihara et al., 2017). Our results further suggest that Col-I density enrichment may not universally lead to YAP activation.

Moreover, Col-I density does not modulate SAV1 and WBP2 expression, respectively inhibitor, and inducer of YAP activity. Our results suggest YAP-independent mechanotransduction in a model of Invasive Ductal Carcinoma similar to recently reported for Ductal Carcinoma in situ (Lee et al., 2019). Therefore care should be taken to evaluate the clinical relevance of YAP-targeting therapies for breast cancer patients (Liu et al., 2017).

Functionally, BCSCs are capable of generating mammospheres in vitro. Culture within Col-I gels is known to not modulate mammosphere formation efficiency relative to the 2D surface (Reynolds et al., 2017). Here we provide evidence that such an effect seems to be independent of Col-I density. On the other hand, unaltered Ki67 levels help to explain the absence of clonogenicity modulation by Col-I density. In terms of morphology, high Col-I density impaired cell elongation, presumably due to smaller pore size caused by increased fiber density. Interestingly, this did not led to decreased fiber alignment and went in line with a previous report showing MDA-MB-231 similarly displaces fibers of low and high Col-I density gels (Riching et al., 2014). Of note, our study was limited in the sense that it only explored Col-I remodeling on Col-I floating gels. Gel restraining would lead to increased sense of stiffness by cells regardless of Col-I density and likely to a pronounced mechanotransduction resulting in greater cellular contractily and matrix remodeling (Wozniak et al., 2003).

Conclusions

In this study, we hypothesized that high Col-I density upregulates BCSC phenotype. On contrary, apart from upregulation of a BCSC immunophenotype, our results suggest that high Col-I density fails to fully impact the BCSC phenotype both in molecular and functional levels. Indeed, more recent reports underscore the importance of tumor microenvironment (Shea et al., 2018) and Col-I topography (Ray et al., 2017) to the maintenance of BCSC phenotype and functionality in Col-I-enriched environments. Future studies should evaluate whether Col-I density impacts stemness of primary cancer cells, which are more prone to plasticity. Considering how the increase of tumor stiffness associated with Col-I enrichment is attracting therapeutic attention (Lampi & Reinhart-King, 2018), our results further support directing efforts into better understanding how matrix topography and composition associates with Col-I-enrichment to drive breast cancer progression and stemness.

Supplemental Information

Supplemental Information 1 Gating strategy for flow cytometry.

FCS-A vs. SSC-A (A) and FCS-A vs. FCS-W (B) dot plots were used to exclude cell debris and cell clumps, respectively. FITC vs. PE dot plots of gated viable singlets were then generated (C) and percentage of population of interest (FITC+/PE-) was obtained (D).

Click here for additional data file.

Supplemental Information 2 Representative morphology of cells within Col-I gels.

Col-I fibers were imaged by confocal reflectance (grey) while cells were stained with CellTracker Green (green) right before image acquisition. Representative cell morphology for each of the Col-I densities is presented. Scale bar: 25 μm.

Click here for additional data file.

Supplemental Information 3 List of gene array fold change regulation relative to 2D group.

Click here for additional data file.

Supplemental Information 4 List of primers used in this study.

Click here for additional data file.

Supplemental Information 5 List of antibodies used in this study.

Click here for additional data file.

We want to thank Professor Rodrigo A. Panepucci (Fundação Hemocentro de Ribeirão Preto, FUNDHERP, Brazil) for kindly donating NTERA-2 cell line. We are also indebted to Mario Costa Cruz (CEFAP, USP) for his assistance with microscopy imaging.

Additional Information and Declarations

Competing Interests

Author Contributions

Data Availability

The authors declare that they have no competing interests.

Iuri C. Valadão conceived and designed the experiments, performed the experiments, analyzed the data, prepared figures and/or tables, authored or reviewed drafts of the paper, and approved the final draft.

Ana Carolina L. Ralph conceived and designed the experiments, performed the experiments, analyzed the data, authored or reviewed drafts of the paper, and approved the final draft.

François Bordeleau conceived and designed the experiments, performed the experiments, analyzed the data, authored or reviewed drafts of the paper, and approved the final draft.

Luciana M. Dzik conceived and designed the experiments, authored or reviewed drafts of the paper, and approved the final draft.

Karen S.C. Borbely conceived and designed the experiments, analyzed the data, authored or reviewed drafts of the paper, and approved the final draft.

Murilo V. Geraldo conceived and designed the experiments, performed the experiments, authored or reviewed drafts of the paper, and approved the final draft.

Cynthia A. Reinhart-King conceived and designed the experiments, authored or reviewed drafts of the paper, and approved the final draft.

Vanessa M. Freitas conceived and designed the experiments, analyzed the data, prepared figures and/or tables, authored or reviewed drafts of the paper, and approved the final draft.

The following information was supplied regarding data availability:

The raw data is available at Figshare: Cordeiro Valadao, Iuri; Ralph, Ana Carolina Lima; Bordeleau, François; Dzik, Luciana Machado; Borbely, Karen Steponavicius Cruz; Geraldo, Murilo Vieira; et al. (2020): Raw data - Iuri et al PeerJ 2020.rar. figshare. Dataset. DOI 10.6084/m9.figshare.11821929.v1.

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
