# Peer review of "High type I collagen density fails to increase breast cancer stem cell phenotype"

_PeerJ, doi:10.7717/peerj.9153_

## Round 0.1 · original submission · Major Revisions

Please revise the manuscript as suggested by both reviewers.

Reviewer 1 ·

Basic reporting

Overall, this paper was written with clear English and enough background. However, the points following should be improved.

1. Where is the statement about Figure 1 in the main text? Please include it.
2. Figure 7: You should include low images for the analysis.

Experimental design

Questions, hypothesis and approach were appropriate. However, several points about experimental design, statement, and drawing in the text should be improved.

1. Lines 90-105 and Figure 1: In Figure 1, the gels look floating. Did you use floating gels or attached gels on the dish in your experiments? You should state which gels you used and draw the clear figures.

2. Lines 174-182, 226-234: You should mention how long you cultured the cells on 2D or in 3D conditions “before” you re-seed the cells on the 2D dishes/wells.

3. Figure 4A: You should clearly divide the upper blot set and the lower blot set for avoiding the confusion. I can see very “narrow” space between the blot of YAP and b-actin, however, it is tough to find the space at the first glance. Some reader may think there are two “b-actin” controls in the same blot set.

Validity of the findings

Most of the data and explanation looks fine, however, several points should be improved.

1. Lines 110-112: You used collagenases for digesting collagen gels. Did you investigate the effects of collagenases on the expression of surface proteins, such as CD44? If no, you should perform an experiment to compare the expression of CD44 in the breast cancer cells on 2D dish with or without collagenase treatment.

2. Lines 147: You used only GAPDH for the housekeeping control for qPCR. You should use at least two housekeeping control targets (e.g., b-actin, a-tubulin, S18) for the experiment because if GAPDH is regulated by collagen density, GAPDH is not a good control.

3. Lines 301-302: There are possibility that not 3D condition but 2D condition increased CSC-related genes in breast cancer cells. On the other hand, collagen itself may be critical for the expression of CSC-related genes. To investigate whether collagen itself is critical for the expression of CSC-related genes in your experimental system, you should perform the experiment to compare CSC-related genes in breast cancer cells on 2D culture plates with or without collagen I coating.

4. Lines 358-377: Why you investigated the expression of E-cadherin and ERalpha? You should mention the reason and background why you included these data in this paper before showing the results.

5. Figure 4, Figure 10: You should use at least two housekeeping control targets (e.g. GAPDH, a-tubulin, histone-H3).

6. In Discussion part, generally the conclusions are stated too much without enough evidence. You should use “may” “indicate” “suggest” for the speculation (for example, lines 425-427).

7. Lines 388-389: In your data in Figure 2, in MDA-MB-231 cells, CD44+CD24- cells were increased in HD condition. You should state the results correctly.

Additional comments

The authors investigated whether high collagen I density affects the stemness in breast cancer cells. Although most data was not positive thus the authors concluded that there is no difference of stemness between breast cancer cells in high and low density collagen gels, most of the experimental data was reliable. However, several points about statement, explanation of the experiments, and data itself were not sufficient. In addition, in Discussion part, several statements looks too much. These points should be improved.

Reviewer 2 ·

Basic reporting

In the manuscript entitled: “High type I collagen density fails to increase breast cancer stem cell phenotype”, the authors carefully investigate the stemness phenotype of two human breast cancer cell lines, MDA-MB-231 and MCF-7 cells, in response to culture in increasing collagen density. While the authors conclude that collagen density is not sufficient to fully develop the BCSC phenotype, the work is rigorous, the manuscript is well written and worthy of publication in PeerJ.

Experimental design

Minor points:
1. The graphic representation of culture conditions was very helpful. With respect to collagen gel culture, the methods section states “Col-I final mixture (175μl) was added to wells of 24-well culture plates (Corning, USA) and placed in the CO2 incubator for 1 hour to polymerize. Col-I gels were then covered with the appropriate cell culture media.”. To this reviewer, that means that the gels remained attached to the bottom of the 24-well plate, which is inconsistent with the graphic in Figure 1. Please reconcile figure 1 with the written methods.

2. As a follow up to comment #1, the authors point out “Col-I gel architecture is affected by factors such as cell culture media, gel thickness, and cell presence, while cell morphology also changes in response to Col-I density”. Restraining the collagen gels by leaving them attached to the bottom of the dish can also influence cellular mechanical signaling and collagen reorganization in response to cellular contractility (Wozniak et al., JCB 2003). Thus, if the gels are attached to the bottom of the dish it is possible that the cells sense a stiff matrix under all collagen concentrations. This may partially explain the lack of collagen fiber reorganization. Perhaps this point can be added to the discussion.

Validity of the findings

This is an relevant and important area of research that will be a benefit to the literature. It is still worth while to note that collagen density may impact CSC markers in primary cell within the native tumor microenvironment. It is quite possible that these well established cell lines no longer have the plasticity to aquire markers associated with the CSC phenotype in response to density.

---

## Round 0.2 · accepted · Accept

Revised manuscript is ready for publication.

Reviewer 1 ·

Basic reporting

The revised points look clear.

Experimental design

The revision of the maniscript looks enough.

Validity of the findings

The revision looks fine. Although there are no additional experimental data, the authors found previous experimental results from already published papers.

Additional comments

Now it's suitable for publication in this journal.